# A quantum algorithm for the $n$-gluon MHV scattering amplitude

Erik Bashore[1][*], Stefano Moretti[1,2][†], Timea Vitos[1,3][‡]

[1] *Department of Physics and Astronomy, Uppsala University,*

*Box 516, 751 20, Uppsala, Sweden*

[2] *School of Physics and Astronomy, University of Southampton,*

*Highfield, Southampton SO17 1BJ, United Kingdom*

[3] *Institute for Theoretical Physics, ELTE Eötvös Loránd University,*

*Pázmány Péter sétány 1/A, H-1117 Budapest, Hungary*

**Abstract**

We propose a quantum algorithm for computing the $n$-gluon maximally helicity violating (MHV) tree-level scattering amplitude. We revisit a newly proposed method for unitarisation of non-unitary operations and present how this implementation can be used to create quantum gates responsible for the color and kinematic factors of the gluon scattering amplitude. As a proof-of-concept, we detail the full conceptual algorithm that yields the squared amplitude and implement the corresponding building blocks on simulated noiseless quantum circuits for $n = 4$ to analyze its performance. The algorithm is found to perform well with parameter optimizations, suggesting it to be a good candidate for implementing on quantum computers also for higher multiplicities.

**Keywords:** Quantum Computing, High-Energy Physics, Scattering Amplitudes

# Contents

[*]erik.bashore.9027@student.uu.se

[†]stefano.moretti@cern.ch

[‡]timea.vitos@physics.uu.se

# 1 Introduction

With the increasing need for simulations and theoretical computations for particle physics and especially collider physics, the topic of alternative computing approaches is timely and of crucial importance to this field of research, in particular, in preparation for the High-Luminosity Large Hadron Collider (HL-LHC) data collection phase [1]. One of the largest computational challenges for the upcoming data collection and analysis is the vast amount of QCD background, which is indeed expected to be the dominant noise to almost all signal processes involving jets in the final state. The latter are known as multi-jet processes as they are characterized by a large number of well-separated jets measured in the detector. The structure of these processes is well known and straightforward to compute with modern event generators, however, with increasing number of final-state particles, the computational complexity reaches a point at which the computation is no longer practical, ranging between 4 and 7 final-state jets, depending on the exact process type, event generator and available computer power.

The need for using alternative approaches for processes involving a large number of external particles becomes apparent when analyzing the multiplicity dependence of the distributions of computing resources for various processes, as reported in Ref. [2]. As the jet multiplicity increases, the fraction of the time spent on amplitude calculation increases and the total time for the amplitude computation grows factorially. This is also expected by examining the color-decomposition of multi-parton amplitudes [3–6]. In this case, the color factors and the kinematics are factorized in a sum running over permutations of the external particles, hence resulting in effectively a double summation of factorial square number of terms. As the total time for the total computation of the cross section becomes dominated by amplitude calculation, the scaling of the total time will also follow this factorial increase with particle multiplicity.

The current state-of-the-art of computing facilities and tools for the theoretical simulation of particle collisions are based on classical computing with Central Processing Unit (CPU) power. Besides various approaches in reformulating the conventional equations in various approximations,

based on, e.g, color-expansion [7–9], there is an active field of research in improving the computational techniques themselves. One of the advances in this direction is towards expanding these conventional computing techniques to facilitate for Graphics Processing Unit (GPU) usage [10–21]. A second area that current particle physics research is exploring is the use of Machine Learning (ML) based techniques in various steps of the simulation process [22–33]. A third direction in the computational advances is a yet not-so-well-explored possibility for tackling the ever-increasing demand of computing power with Quantum Computing (QC) techniques, which is the approach to be discussed and explored in the current paper.

As the field of quantum computers is still under active research and development, the application of it to particle physics is also at very early stages, however, there are already numerous areas within such a discipline where research has showed interesting results for practical use of QC. These works include application within lattice QCD [34–37], loop computations in Feynman graphs [38–40], effective field theories [41], parton distribution functions [42, 43], parton shower algorithms [44, 45], integration of amplitudes [46, 47] and event generators in general [48–50]. Yet another area is the computation of the hard scattering process itself, which is the leading power consumer for multi-jet processes. For the hard process part of the event simulation, some attempts have been made [51, 52]. Among these, we highlight here two main works. The first is Ref. [53], where the possibility of applying QC to the evaluation of color factors in QCD amplitudes is presented. The second is Ref. [44], in which the scope of applying QC to helicity-amplitude evaluation is presented.

In this work, we investigate the possibility of using QC for the calculation of the full amplitude for all-gluon processes, through an algorithm which combines both the color-factor and helicity-amplitude computations, for maximally helicity-violating (MHV) configurations, following closely the conventions of Refs. [54] and [44]. A similar approach was recently adopted in Ref. [55], which allows for the computation of color amplitudes (and therefore interferences) instead of squared amplitudes as in Ref. [53].

The paper is organized as follows. In Sec. 2 we give an overview of the conventions used: specifically, in Sec. 2.1 we give a brief introduction to QCD amplitudes while in Sec. 2.2 we present the constructions of the helicity- and color-factor gates. Then, in Sec. 3, we present the full algorithm for the computation of the amplitudes. In Sec. 4 we present the analysis and results. We finally summarize as well as present our concluding remarks and outlook in Sec. 5.

## 2 Background and conventions

### 2.1 Scattering amplitudes in QCD

In the high-energy realm of particle collisions ($Q^2 \gg \Lambda_{\mathrm{QCD}}$), the QCD coupling is small enough to perform perturbative QCD calculations. A QCD matrix element of any process is expanded, in its most schematically simplistic form, as

$$\mathcal{M} = \sum_\sigma \mathcal{C}_\sigma \mathcal{A}_\sigma \tag{1}$$

where $\mathcal{C}_\sigma$ are color factors, originating from the strong interaction vertices, and $\mathcal{A}_\sigma$ are momentum-dependent amplitudes, originating from wave functions and propagators. Here, the $\sigma$ indices represent generic permutations of the external particles. The expansion is not unique, though, as there exist widely used bases, which are convenient for various purposes [56–58]. In the following,

however, we focus on the all-gluon process at leading-order (LO) accuracy in QCD perturbation theory, in the fundamental representation [3], given by

$$\mathcal{M} = g^{n-2} \sum_{\sigma \in S_{n-1}} \text{Tr}\Big[T^{a_1}\sigma\big(T^{a_2}, \ldots, T^{a_n}\big)\Big] \mathcal{A}(1, \sigma(2, ..., n)). \tag{2}$$

As common procedure for the unpolarized hadronic collisions, the amplitude is computed with a summation over all possible helicity states of the external particles (and average over initial ones). It is well-known, however, that not all helicity configurations contribute to the total sum with equal weight, some of these being completely vanishing and some being negligibly small [4, 59, 60]. The dominant contributions are the MHV amplitudes, which were named so in the light of the expectation that helicity is to be conserved in (massless) QCD collisions. These amplitudes (also known as Parke-Taylor amplitudes) were shown to have a very simple form for $n$-gluon amplitudes, namely:

$$\mathcal{A}(1^-, 2^+, 3^+ \ldots k^-, \ldots, n^+) = \frac{\langle 1k \rangle^4}{\langle 12 \rangle \langle 23 \rangle \ldots \langle n1 \rangle}, \tag{3}$$

where the angular brackets denote the spinor inner products in the helicity formalism [3]. We choose to work in the commonly used convention that the helicities of the 1st and the $k$th particles are set to minus, while all others have plus values.

While the next-to-leading-MHV amplitudes have been worked out [60], this simple structure does not appear therein, yet, for many purposes, the MHV amplitudes are still used as approximations in various forms [9]. Upon squaring the matrix element, the interference terms between the different helicity configurations vanish, effectively leading to a summation over squares of different helicity configurations. In this work, the MHV amplitudes are considered, being indeed the dominant contributions to the all-gluon amplitude.

## 2.2 Quantum gates

In this section we address the two basic quantum gates needed in order to build the general algorithm for the computation of the full amplitude. These gates are the color-factor gate $U_{\mathcal{C}}$ and the helicity gate $U_{\mathcal{A}}$, respectively. A key component used for both of these is the method of unitarisation of non-unitary operations which is described in Ref. [53] and reviewed once more in Appendix A.

### 2.2.1 Color-factor gate

First, we consider the color-factor part of the expansion in Eq. (2). The construction of the color-factor gate $U_{\mathcal{C}}$ follows closely the notation and steps used in Ref. [53]. The main objective is to create the abstract action

$$|a_1 a_2 ... a_n\rangle_{\{g_i\}} \mapsto \text{Tr}\Big[T^{a_1} T^{a_2} ... T^{a_n}\Big] |a_1 a_2 ... a_n\rangle_{\{g_i\}} \tag{4}$$

where $|a_1 a_2 ... a_n\rangle_{\{g_i\}}$ is an $n$-gluon reference color-state where each gluon register $g_i$ is composed of 3 qubits and the encoding of the 8 gluon colors is the binary representation of integers

$$|a\rangle_g \in \big\{|1\rangle_g, ..., |8\rangle_g\big\} = \big\{|000\rangle, ..., |111\rangle\big\}. \tag{5}$$

To perform this action one needs ancilla registers $q\bar{q}$ and $\mathcal{U}$ which constitutes 4 and $n_u$ qubits each as described in Ref. [53]. From this one can define the $Q$ gate that acts on the $n = 1$ case as

$$Q\left(|a\rangle_g \otimes |k\rangle_q \otimes |\Omega\rangle_\mathcal{U}\right) = \sum_{j=1}^{3} T_{jk}^a |a\rangle_g |j\rangle_q |\Omega\rangle_\mathcal{U} + \left(\perp |\Omega\rangle_\mathcal{U}\right) \tag{6}$$

where $|k\rangle_q \in \{|1\rangle_q, |2\rangle_q, |3\rangle_q\} = \{|00\rangle, |01\rangle, |10\rangle\}$ represents the fundamental indices of the generators and $|\Omega\rangle_\mathcal{U} = |0\rangle^{\otimes n_u}$ is a unitarity reference state which ensures that all relevant factors are placed in front of this reference with the rest being orthogonal to it. Considering now a 4-qubit rotation gate $R_{q\bar{q}}$ (see Appendix B for its construction) that acts on the $q\bar{q}$ vacuum state as

$$R_{q\bar{q}}|\Omega\Omega\rangle_{q\bar{q}} = \frac{1}{\sqrt{3}} \sum_{k=1}^{3} |kk\rangle_{q\bar{q}} \tag{7}$$

as well as the $U_\mathcal{C}$ gate which we define to be

$$U_\mathcal{C} \equiv \bigotimes_{j=n}^{1} Q^{(j)} \tag{8}$$

where $Q^{(j)}$ acts only on the $j^{\text{th}}$ gluon register. The circuit representation of $U_\mathcal{C}$ can be seen in Fig. 1. The composition $R_{q\bar{q}}^\dagger U_\mathcal{C} R_{q\bar{q}}$ acts then on the initial state

$$|\psi_0\rangle = |a_1 a_2 ... a_n\rangle_{\{g_i\}} \otimes |\Omega\Omega\rangle_{q\bar{q}} \otimes |\Omega\rangle_\mathcal{U} \tag{9}$$

in the following way, using Eq. (7) and Eq. (6),

$$R_{q\bar{q}}^\dagger U_\mathcal{C} R_{q\bar{q}}|\psi_0\rangle = R_{q\bar{q}}^\dagger \bigotimes_{j=n}^{1} Q^{(j)} \frac{1}{\sqrt{3}} \sum_{k=1}^{3} |a_1 a_2 ... a_n\rangle_{\{g_i\}} |kk\rangle_{q\bar{q}} |\Omega\rangle_\mathcal{U}$$

$$= \frac{1}{\sqrt{3}} R_{q\bar{q}}^\dagger Q^{(1)} Q^{(2)} ... Q^{(n-1)} \sum_{k,l_1}^{3} T_{l_1 k}^{a_n} |a_1 a_2 ... a_n\rangle_{\{g_i\}} |l_1 k\rangle_{q\bar{q}} |\Omega\rangle_\mathcal{U} + \left(\perp |\Omega\rangle_\mathcal{U}\right)$$

$$= \frac{1}{\sqrt{3}} R_{q\bar{q}}^\dagger \sum_{k,\{l_i\}} T_{l_n l_{n-1}}^{a_1} T_{l_{n-1} l_{n-2}}^{a_2} ... T_{l_1 k}^{a_n} |a_1 a_2 ... a_n\rangle_{\{g_i\}} |l_n k\rangle_{q\bar{q}} |\Omega\rangle_\mathcal{U} + \left(\perp |\Omega\rangle_\mathcal{U}\right)$$

$$= \frac{1}{3} \text{Tr}\left[T^{a_1} T^{a_2} ... T^{a_n}\right] |a_1 a_2 ... a_n\rangle_{\{g_i\}} |\Omega\Omega\rangle_{q\bar{q}} |\Omega\rangle_\mathcal{U} + \left(\perp |\Omega\Omega\rangle_{q\bar{q}} |\Omega\rangle_\mathcal{U}\right)$$

where the first $R_{q\bar{q}}$ opens the superposition of the $q\bar{q}$ register and each $Q^{(j)}$ of $U_\mathcal{C}$ pulls out the corresponding $T_{l_{n-j+1} l_{n-j}}^{a_j}$ factor and contracts the left index with the right index of the previous generator where the final $R_{q\bar{q}}^\dagger$ sends $l_n$ to $k$ so as to close the trace yielding the final line. Note now how the complete action takes the reference gluon-color state $|a_1 a_2 ... a_n\rangle$ and places its corresponding color-factor $\text{Tr}\left[T^{a_1} T^{a_2} ... T^{a_n}\right]$ as a probability amplitude in front with the total state being normalized by the help of the $\mathcal{U}$ register.

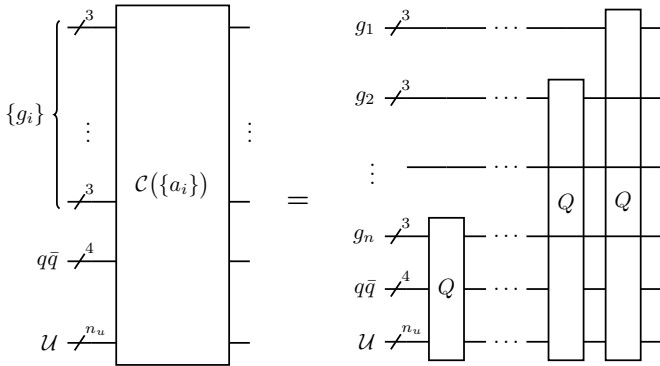

Figure 1: Circuit decomposition of the $U_{\mathcal{C}}$ gate defined in Eq. (8).

### 2.2.2 Helicity-amplitude gate

Next, the dual amplitudes are considered from Eq. (2), for which the $U_{\mathcal{A}}$ gate is introduced, providing with $U_{\mathcal{A}}|\psi_{\text{ref}}\rangle \mapsto \mathcal{A}|\psi_{\text{ref}}\rangle$ for some reference state $|\psi_{\text{ref}}\rangle$, where $\mathcal{A}$ is given by the Parke-Taylor formula in Eq. (3). The procedure is to setup a set of $n-1$ momentum registers $\{k_i\}$ that determine the ordering in $\mathcal{A}(1, k_2, ..., k_n)$. We need to encode the momentum labels for this and so each $k_i$ register contains $n_k = \lceil \log_2(n-1) \rceil$ qubits. Recall that in the full amplitude the first leg is stationary in the permutations and thus we do not need to encode its position. For this gate we will have a unitary register $\mathcal{U}$ where the value setting gates $B(\alpha)$ (defined in Appendix B) will be controlled by the $\{k_i\}$ registers. In line with Eq. (3) we first want to find which $k_2$ is contracted with spinor 1 in the product $\langle 1k_2\rangle^{-1}$ and so we first append controlled value setting gates steered by the state $|i\rangle$ of register $k_2$, i.e., using the operator $C_{|i\rangle}^{(k_2)}\big[B(\langle 1i\rangle^{-1})\big]$. Afterwards, we want to add the string of spinor products $\big(\langle k_2 k_3\rangle\langle k_3 k_4\rangle...\langle k_{n-1}k_n\rangle\big)^{-1}$ as well as the final piece $\langle k_n 1\rangle^{-1}$, which we do with the operators $\bigotimes_{a=2}^{n-1} C_{|ij\rangle}^{(k_a \otimes k_{a+1})}\big[B(\langle ij\rangle^{-1})\big]$ and $C_{|i\rangle}^{(k_n)}\big[B(\langle i1\rangle^{-1})\big]$, respectively. We end by applying the numerator $\langle 1\lambda\rangle^4$ where we label the other anomalous helicity by $\lambda$. This is done with a helicity register $h$ which controls the value of $\lambda$ and the gate we need to append is $C_{|\lambda\rangle}^{(h)}\big[B(\langle 1\lambda\rangle^4)\big]$. Putting all of these operators together with increment operator $U_+$ squeezed in-between each, we find our full partial amplitude gate to be

$$U_{\mathcal{A}} \equiv \left( C_{|\lambda\rangle}^{(h)}\big[B(\langle 1\lambda\rangle^4)\big]U_+ \right) \otimes \left( C_{|i\rangle}^{(k_n)}\big[B(\langle i1\rangle^{-1})\big]U_+ \right)$$
$$\bigotimes_{a=2}^{n-1} \left( C_{|ij\rangle}^{(k_a \otimes k_{a+1})}\big[B(\langle ij\rangle^{-1})\big]U_+ \right) \otimes \left( C_{|i\rangle}^{(k_2)}\big[B(\langle 1i\rangle^{-1})\big]U_+ \right). \tag{10}$$

The full circuit diagram for this gate can be seen in Fig. 2 and from this it should be noted that the action on some arbitrary reference state $|\lambda\rangle_h|k_2 k_3...k_n\rangle_{\{k_i\}}|\Omega\rangle_{\mathcal{U}}$ produces

$$U_{\mathcal{A}}\left( |\lambda\rangle_h|k_2 k_3...k_n\rangle_{\{k_i\}}|\Omega\rangle_{\mathcal{U}} \right) = \frac{\langle 1\lambda\rangle^4}{\langle 1k_2\rangle\langle k_2 k_3\rangle...\langle k_n 1\rangle}|\lambda\rangle_h|k_2 k_3...k_n\rangle_{\{k_i\}}|\Omega\rangle_{\mathcal{U}} + \left( \perp |\Omega\rangle_{\mathcal{U}} \right). \tag{11}$$

Note, however, that for the insertion of $\langle ij \rangle^{-1}$ into the value setting gates $B(\alpha)$, it might be the case that $|\langle ij \rangle^{-1}| > 1$ rendering $B(\alpha)$ non-unitary. We solve this problem by introducing a real parameter $\varepsilon$ which obeys

$$\varepsilon \geq |\langle ij \rangle^{-1}| \quad \forall i, j \tag{12}$$

so that if we divide it over all input values $\alpha$, we find

$$0 < \left| \frac{1}{\varepsilon \langle ij \rangle} \right| \leq 1 \quad \forall i, j, \tag{13}$$

yielding unitary $B(\alpha)$ operators. We then end up with an overall factor of $\varepsilon^{-n}$ in the final output which we simply can multiply out in post-processing. We do not need to do this for the numerator as all $\langle 1\lambda \rangle^4$ have norm less than 1. This input of $\varepsilon$ is suppressed in the definitions and diagrams but should not be forgotten in the implementation.

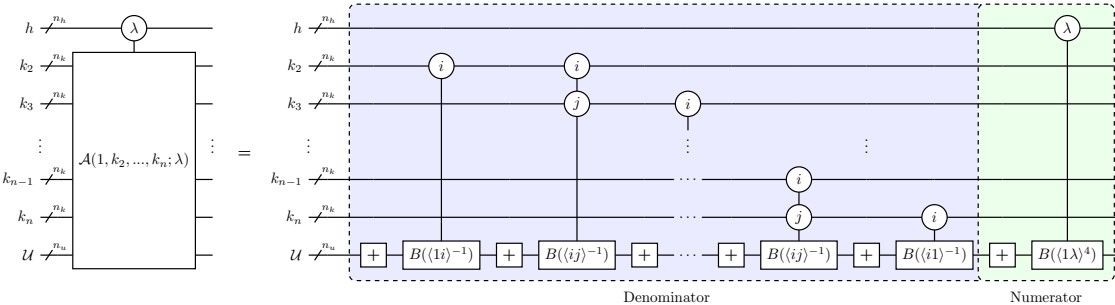

Figure 2: Circuit decomposition of the $U_{\mathcal{A}}$ gate defined in Eq. (10).

# 3 The algorithm

With the introduced gates $U_{\mathcal{C}}$ and $U_{\mathcal{A}}$ we now propose a method to fully compute the $n$-gluon MHV scattering amplitude in a quantum circuit. By computing the amplitude, we mean to initialize a quantum state on a set of registers and then utilize unitary operations so that the amplitude defined in Eq. (2) becomes the probability amplitude corresponding to some reference state. Note that for a given $n$, the amplitude is given by a sum over $(n-1)!$ permutations $\sigma \in S_{n-1}$. Our first goal is then to initialize a *permutation register* $p$ whose states encode the set of permutation labels:

$$|\sigma\rangle_p \in \left\{ |1\rangle, |2\rangle, ..., |(n-1)!\rangle \right\}, \tag{14}$$

which can be done with a register of $\lceil \log_2[(n-1)!] \rceil$ qubits. Then we initialize that register as a normalized superposition of all relevant permutations, i.e., the encoded states

$$|\Sigma\rangle_p \sim \sum_\sigma |\sigma\rangle_p. \tag{15}$$

With the quantum state prepared, we would like to combine it with ancilla registers and use the gates $U_\mathcal{C}$ and $U_\mathcal{A}$ such that we produce the following action (up to a normalization)

$$|\Sigma\rangle_p \mapsto \sum_\sigma \mathcal{C}_\sigma \mathcal{A}_\sigma |\sigma\rangle_p, \tag{16}$$

where $\mathcal{C}_\sigma$ and $\mathcal{A}_\sigma$ are the color- and helicity-factors corresponding to the given permutation, respectively. If we were to create this state, then all we need to do is apply the Quantum Fourier Transform (QFT) to the permutation register, as then all independent amplitudes decouple from the permutation states and the pure sum over all terms ends up in front of the vacuum state as follows:

$$\widehat{\mathrm{QFT}}\left( \sum_\sigma \mathcal{C}_\sigma \mathcal{A}_\sigma |\sigma\rangle_p \right) \sim \left( \sum_\sigma \mathcal{C}_\sigma \mathcal{A}_\sigma \right)|\Omega\rangle_p + \left( \perp |\Omega\rangle_p \right), \tag{17}$$

again, valid up to some normalization. The output given by measuring the state would then automatically yield the absolute value squared of the full amplitude $\left| \sum_\sigma \mathcal{C}_\sigma \mathcal{A}_\sigma \right|^2$ enabling us to instantly compute *all interference terms in one measurement*. The reason why Eq. (17) works in that particular fashion can be seen from the definition of the QFT. Recall that the action of the QFT on an arbitrary quantum state $|\psi\rangle = \sum_l \alpha_l |l\rangle$ is

$$\widehat{\mathrm{QFT}}|\psi\rangle = \frac{1}{\sqrt{N}} \sum_{j=0}^{N-1} \sum_{l=0}^{N-1} e^{2\pi i (jl/N)} \alpha_l |j\rangle, \tag{18}$$

where we note that, for $j = 0$, all exponents in the QFT coefficients become 1 and we have

$$\widehat{\mathrm{QFT}}|\psi\rangle = \frac{1}{\sqrt{N}} \left( \sum_{l=0}^{N-1} \alpha_l |0\rangle + \sum_{j=1}^{N-1} \sum_{l=0}^{N-1} e^{2\pi i (jl/N)} \alpha_l |j\rangle \right). \tag{19}$$

We see that the pure sum without any phase-factors is the coefficient in front of the vacuum state which is exactly what happens in Eq. (17).

The last step is to just include all different color and helicity configurations, which we can simply do with a Hadamard and other superposition gates to then, in post-processing, add together all contributions to acquire

$$\sum_{\substack{\text{color} \\ \text{helicity}}} \left| \sum_\sigma \mathcal{C}_\sigma \mathcal{A}_\sigma \right|^2. \tag{20}$$

This is the underlying idea of how the algorithm works. Now we move to a walk-through of the algorithm step-by-step, see Fig. 3.

**Step 0: Initialize registers**. The circuit for the full algorithm consists of a helicity register $h$, a permutation register $p$, a set of $n-1$ momentum registers $\{k_i\}$, a set of $n$ gluon registers $\{g_i\}$, a quark/anti-quark register $q\bar{q}$ and a unitarity register $\mathcal{U}$. The number of qubits needed for each individual register and the total number of qubits is detailed in the next section. The initial state on the quantum computer is thus

$$|\psi_{\text{init}}\rangle = |\Omega\rangle_h \otimes |\Omega\rangle_p \bigotimes_{j=2}^{n} |\Omega\rangle_{k_j} \bigotimes_{m=1}^{n} |\Omega\rangle_{g_m} \otimes |\Omega\rangle_{q\bar{q}} \otimes |\Omega\rangle_\mathcal{U}, \tag{21}$$

where $\Omega$ indicates the "vacuum" state and is synonymous with $|\Omega\rangle_r = |0\rangle^{\otimes n_r}$ for register $r$.

**Step 1.1: Prepare state**. Next, the superposition gates $R_\lambda, R_\sigma, R_{q\bar{q}}$ and Hadamard gates are appended that open up superpositions in registers $h$, $p$, $q\bar{q}$, $\{g_i\}$ and also the initialization of the momentum register $\{k_i\}$ with corresponding $X$ gates from $R_k$. The prepared state is given by (again, omitting normalization factors for now)

$$|\psi'_{\text{init}}\rangle \equiv \left( R_\lambda \otimes R_\sigma \otimes R_k \bigotimes_{m=1}^{n} H^{(g_m)} \otimes R_{q\bar{q}} \right) |\psi_{\text{init}}\rangle$$

$$\sim \sum_{\lambda,\sigma} |\lambda\rangle_h |\sigma\rangle_p |23...n\rangle_{\{k_i\}} \sum_{\{a_i\}} |a_1 a_2 ... a_n\rangle_{\{g_i\}} \sum_{k=1}^{3} |kk\rangle_{q\bar{q}} |\Omega\rangle_\mathcal{U}. \tag{22}$$

**Step 1.2: Controlled SWAPs**. Apply controlled SWAP gates that read the permutation state and swaps the ordering of the momentum registers $\{k_i\}$ and gluon registers $\{g_i\}$ correspondingly. The gates are named $g$SWAP and $k$SWAP respectively and are generalizations of the well-known Fredkin gate. An example of $g$SWAP gate can be found in Appendix B. The prepared state then reads

$$|\psi_1\rangle \equiv \left( k\text{SWAP} \right)\left( g\text{SWAP} \right)|\psi'_{\text{init}}\rangle$$

$$\sim \sum_{\lambda,\sigma} |\lambda\rangle_h |\sigma\rangle_p |\sigma(2,3,...,n)\rangle_{\{k_i\}} \sum_{\{a_i\}} |a_1\sigma(a_2,...,a_n)\rangle_{\{g_i\}} \sum_{k=1}^{3} |kk\rangle_{q\bar{q}} |\Omega\rangle_\mathcal{U}. \tag{23}$$

**Step 2: Compute color/helicity factors**. Computation gates $U_\mathcal{C}$ and $U_\mathcal{A}$ are applied that read the momentum and gluon register and apply the corresponding helicity- and color-factors, respectively. The altered state reads

$$|\psi_2\rangle \equiv \left( U_\mathcal{A} \otimes U_\mathcal{C} \right)|\psi_1\rangle$$

$$\sim \sum_{\lambda,\sigma} \mathcal{A}(1,\sigma(2,3,...,n);\lambda) \sum_{\{a_i\}} \sum_{k,\{l_i\}} T^{a_1}_{l_n l_{n-1}} T^{\sigma(a_2)}_{l_{n-1} l_{n-2}} ... T^{\sigma(a_n)}_{l_1 k} |\lambda\rangle_h |\sigma\rangle_p \tag{24}$$

$$\otimes |\sigma(2,3,...,n)\rangle_{\{k_i\}} |a_1\sigma(a_2,...,a_n)\rangle_{\{g_i\}} |l_n k\rangle_{q\bar{q}} |\Omega\rangle_\mathcal{U} + \left( \perp |\Omega\rangle_\mathcal{U} \right).$$

**Step 3.1: Inverse controlled SWAPs**. Reverse the permutation of the momentum and gluon registers by applying inverse controlled SWAP gates, this collects the sum of permutation-dependent amplitudes in front of the original state and the output is

$$|\psi'_2\rangle \equiv \left( k\text{SWAP}^\dagger \right)\left( g\text{SWAP}^\dagger \right)|\psi_2\rangle$$

$$\sim \sum_{\lambda,\sigma} \mathcal{A}(1,2,3,...,n;\lambda) \sum_{\{a_i\}} \sum_{k,\{l_i\}} T^{a_1}_{l_n l_{n-1}} T^{\sigma(a_2)}_{l_{n-1} l_{n-2}} ... T^{\sigma(a_n)}_{l_1 k} |\lambda\rangle_h |\sigma\rangle_p \tag{25}$$

$$\otimes |2,3,...,n\rangle_{\{k_i\}} |a_1 a_2,...,a_n\rangle_{\{g_i\}} |l_n k\rangle_{q\bar{q}} |\Omega\rangle_\mathcal{U} + \left( \perp |\Omega\rangle_\mathcal{U} \right).$$

**Step 3.2: Close trace and reset momentum registers**. Apply an inverse $q\bar{q}$ rotation gate $R^\dagger_{q\bar{q}}$ to close the trace and also reset the momentum registers back to the vacuum state with $R^\dagger_k$, the

organized output is

$$
\begin{aligned}
|\psi_3\rangle &\equiv \left(R_k^\dagger \otimes R_{q\bar{q}}^\dagger\right)|\psi_2'\rangle \\
&\sim \sum_{\lambda,\sigma}\sum_{\{a_i\}}\mathrm{Tr}\big[T^{a_1}T^{\sigma(a_2)}...T^{\sigma(a_n)}\big]\mathcal{A}(1,\sigma(2,3,...n);\lambda) \\
&\times |\lambda\rangle_h|\sigma\rangle_p|\Omega\rangle_{\{k_i\}}|a_1a_2...a_n\rangle_{\{g_i\}}|\Omega\rangle_{q\bar{q}}|\Omega\rangle_{\mathcal{U}} + \left(\perp |\Omega\rangle_{q\bar{q}}|\Omega\rangle_{\mathcal{U}}\right).
\end{aligned}
\tag{26}
$$

**Step 4: QFT the permutation register**. The final step of the algorithm is collecting the full sum of permutations in front of the vacuum state of the permutation register and then computing the interference of all terms. This is done with the QFT operator on the permutation register that leaves the pure sum in front of $|\Omega\rangle_p$ and leads to the final state

$$
\begin{aligned}
|\psi_{\mathrm{final}}\rangle &\equiv \widehat{\mathrm{QFT}}_{(p)}|\psi_3\rangle \\
&= \frac{1}{\sqrt{\mathcal{N}}}\sum_{\lambda,\{a_i\}}\left(\sum_\sigma \mathrm{Tr}\big[T^{a_1}T^{\sigma(a_2)}...T^{\sigma(a_n)}\big]\mathcal{A}(1,\sigma(2,3,...n);\lambda)\right)|\lambda\rangle_h|a_1a_2...a_n\rangle_{\{g_i\}} \\
&\otimes |\Omega\rangle_p|\Omega\rangle_{\{k_i\}}|\Omega\rangle_{q\bar{q}}|\Omega\rangle_{\mathcal{U}} + \left(\perp |\Omega\rangle_p|\Omega\rangle_{q\bar{q}}|\Omega\rangle_{\mathcal{U}}\right).
\end{aligned}
\tag{27}
$$

For the final step we added the combined normalization to get the precise answer with the factor

$$
\mathcal{N} = 9 \cdot 8^n \cdot 2^{n_p} \cdot \varepsilon^{2n} \cdot n_\sigma \cdot n_\lambda.
\tag{28}
$$

In the expression above the state has picked up a factor of 3 from the opening and closing of the $q\bar{q}$ superposition, a factor $8^{n/2}$ from the $\{g_i\}$ superpositions, a factor $2^{n_p}$ from the final QFT, a factor $\varepsilon^{2n}$ from the $U_{\mathcal{A}}$ gate, a factor $n_\sigma = \#$permutations from the initial preparation of the permutation register $p$ and, lastly, a factor $n_\lambda = \#$helicity configurations from the initialization of the helicity register $h$. The complete circuit diagram for all steps 0-4 can be seen in Fig. 3.

**Output:** When measuring the final state one retrieves the absolute value squared of the full color-dressed amplitude

$$
\left|\langle\lambda|\langle a_1a_2...a_n|\psi_{\mathrm{final}}\rangle\right|^2 = \frac{1}{\mathcal{N}}\left|\sum_\sigma \mathrm{Tr}\big[T^{a_1}T^{\sigma(a_2)}...T^{\sigma(a_n)}\big]\mathcal{A}(1,\sigma(2,3,...n);\lambda)\right|^2,
\tag{29}
$$

with helicity and color labels defined up to a normalization. There is of course an implicit projection to the vacuum states of the remaining registers in the statement above. In such a way one receives all the possible color and helicity configurations of $|\mathcal{M}_{\mathrm{Tree}}^{\mathrm{MHV}}|^2$ simultaneously which easily can be summed together in post-processing to acquire

$$
\sum_{\substack{\mathrm{color} \\ \mathrm{helicity}}}\left|\mathcal{M}_{\mathrm{Tree}}^{\mathrm{MHV}}\right|^2 = \sum_{\lambda,\{a_i\}}\mathcal{N}\left|\langle\lambda|\langle a_1a_2...a_n|\psi_{\mathrm{final}}\rangle\right|^2.
\tag{30}
$$

The actual summation above of the probabilities can not be done on the quantum device as the probabilities are gathered when measuring the circuit.

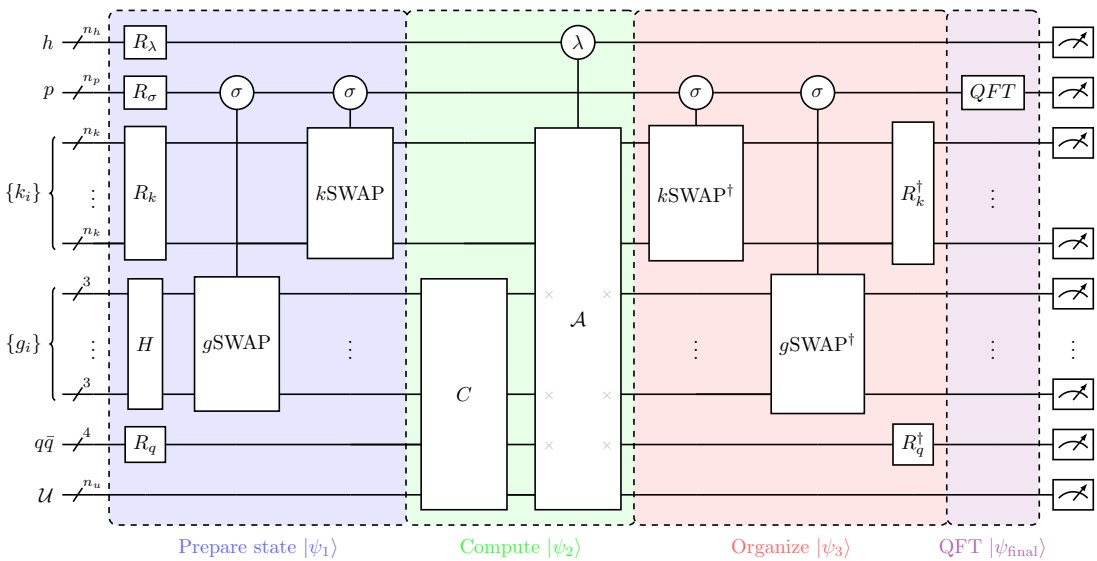

Figure 3: Circuit diagram representation of the algorithm detailed in steps 0-4.

# 4  Results

In order to verify its performance and efficiency we perform multiple tests by running individual gates and sub-procedures to be compared with the expected results from formulae. This gives a sense of how all the individual pieces put together can yield promising results. To do this, we implement the gate in the IBM's python module *Qiskit* [61], from which we either measure the circuit outputs or use the built in 'state-vector method' to efficiently extract the complex amplitudes of the circuit state-vector. For all of these tests we limit ourselves to the 4-gluon scattering amplitude, for which we study a collision along the $z$-axis. The corresponding input helicity angles for this process are listed below in Table 1 (the third variable for each momentum is the energy, which is an overall normalization for the spinors and is hence omitted).

| Spinor $i$ | 1 | 2 | 3 | 4 |
|---|---|---|---|---|
| $\theta_i$ | 0.0 | $\pi$ | 1.326417 | -1.815175 |
| $\varphi_i$ | 0.0 | 0.0 | 1.981338 | 1.160255 |

Table 1: Spinor angles $\theta_i$ and $\varphi_i$ used for the single spinor input tests.

## 4.1  4-point color computation

We test a 4-point color computation with a specific gluon color input $\{1, 2, 4, 5\}$ as an example. The goal is to compute the color-factor $\text{Tr}\left[T^1, T^2, T^4, T^5\right]$ and the 6 color orderings of the gluons. The circuit for this computation contains a 3-qubit permutation register $p$, four gluon registers $\{g_i\}$, a

quark/anti-quark register $q\bar{q}$ and a 3-qubit unitarity register $\mathcal{U}$. The encoding of the 6 permutations is chosen by the integer representation of the labels

$$|\sigma\rangle_p \in \big\{|1\rangle_p, ..., |6\rangle_p\big\} = \big\{|000\rangle, |001\rangle, |010\rangle, |011\rangle, |100\rangle, |101\rangle\big\} \tag{31}$$

and the unitary gate $R_\sigma$ that creates this initial state can be found in the appendix in Eq. (B.5). Using the $g$SWAP gate for $n = 4$, which can be seen in Fig. B.4, together with the color computational gate $U_\mathcal{C}$ the expected output state is

$$\frac{1}{3\sqrt{6}} \sum_\sigma \mathrm{Tr}\big[T^1 \sigma(T^2, T^4, T^5)\big] |1245\rangle_{g_1...g_4} |\sigma\rangle_p |\Omega\rangle_{q\bar{q}} |\Omega\rangle_\mathcal{U} + \bigg( \perp |\Omega\rangle_{q\bar{q}} |\Omega\rangle_\mathcal{U} \bigg), \tag{32}$$

where the normalization comes from the $R_{q\bar{q}}$ and $R_{q\bar{q}}^\dagger$ gates as well as the permutation superposition. The full circuit can be seen in Fig. 4. For this test we are interested in eventual sign changes of the permutations and so we cannot measure the circuit in the usual manner as that would yield an absolute value of each factor where the information of a negative sign would be lost. In order to circumvent this sign problem, the state-vector method was used to extract the relevant amplitudes with signs preserved. This method finds the state-vector of a given quantum circuit as an array of the complex probability amplitudes $\alpha_i$ in $|\psi\rangle = \sum_i \alpha_i |x_i\rangle$. The results are presented in Table 2.

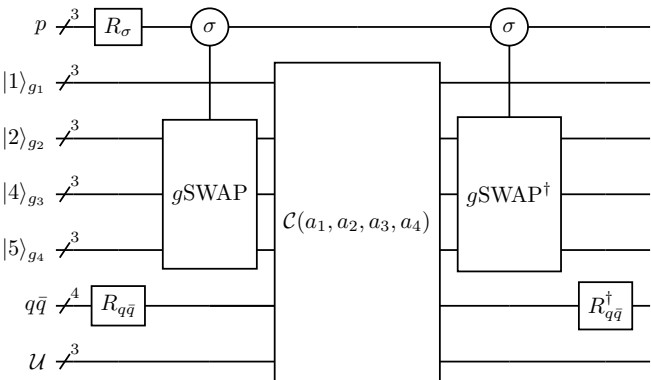

Figure 4: Color-factor computation circuit for the color input $\{1, 2, 4, 5\}$. The $\mathcal{C}$ gate decomposition can be seen in Fig. 1.

## 4.2 Multi-partial amplitude circuit

We test a multi-partial amplitude circuit in the $s$-channel with one permutation register $p$, three momentum registers $k_2, k_3$ and $k_4$ plus one unitarity register $\mathcal{U}$. The circuit is initialized with the momentum registers in the state $|234\rangle_{\{k_i\}} = |01\rangle|10\rangle|11\rangle$ and a superposition of 6 permutations in register $p$ with the $R_\sigma$ gate:

$$R_\sigma \bigg( |\Omega\rangle_p |234\rangle_{\{k_i\}} |\Omega\rangle_\mathcal{U} \bigg) = \frac{1}{\sqrt{6}} \sum_\sigma |\sigma\rangle_p |234\rangle_{\{k_i\}} |\Omega\rangle_\mathcal{U}. \tag{33}$$

| Color factor | Output | True value |
|:---:|:---:|:---:|
| $\mathrm{Tr}\big[T^1T^2T^4T^5\big]$ | -0.0625 | -0.0625 |
| $\mathrm{Tr}\big[T^1T^4T^2T^5\big]$ | 0.0 | 0.0 |
| $\mathrm{Tr}\big[T^1T^2T^5T^4\big]$ | 0.0625 | 0.0625 |
| $\mathrm{Tr}\big[T^1T^5T^4T^2\big]$ | -0.0625 | -0.0625 |
| $\mathrm{Tr}\big[T^1T^4T^5T^2\big]$ | 0.0625 | 0.0625 |
| $\mathrm{Tr}\big[T^1T^5T^2T^4\big]$ | 0.0 | 0.0 |

Table 2: Result from the output state in Eq. (32) after removing normalization factors.

Afterwards we append a $k$SWAP gate, a helicity amplitude gate $U_{\mathcal{A}}$ and an inverse $k$SWAP gate which should generate the output state

$$
\begin{aligned}
|\psi_{\text{final}}\rangle &= k\text{SWAP}^{\dagger}U_{\mathcal{A}}k\text{SWAP}\left(\frac{1}{\sqrt{6}}\sum_{\sigma}|\sigma\rangle_p|234\rangle_{\{k_i\}}|\Omega\rangle_{\mathcal{U}}\right) \\
&= k\text{SWAP}^{\dagger}U_{\mathcal{A}}\left(\frac{1}{\sqrt{6}}\sum_{\sigma}|\sigma\rangle_p|\sigma(234)\rangle_{\{k_i\}}|\Omega\rangle_{\mathcal{U}}\right) \\
&= k\text{SWAP}^{\dagger}\left(\frac{1}{\sqrt{6}}\sum_{\sigma}\mathcal{A}(1^-,\sigma(2^-,3^+,4^+))|\sigma\rangle_p|\sigma(234)\rangle_{\{k_i\}}|\Omega\rangle_{\mathcal{U}}\right)+\left(\perp|\Omega\rangle_{\mathcal{U}}\right) \\
&= \frac{1}{\sqrt{6}}\sum_{\sigma}\mathcal{A}(1^-,\sigma(2^-,3^+,4^+))|\sigma\rangle_p|234\rangle_{\{k_i\}}|\Omega\rangle_{\mathcal{U}}+\left(\perp|\Omega\rangle_{\mathcal{U}}\right).
\end{aligned}
\tag{34}
$$

The diagram for this circuit can be seen in Fig. 5. From this we can find the absolute squared of the amplitudes by running the circuit with $\chi$ shots and compute

$$
\big|\langle\sigma|\psi_{\text{final}}\rangle\big|^2 = \frac{1}{6}\big|\mathcal{A}(1^-,\sigma(2^-,3^+,4^+))\big|^2 \sim \frac{N_{\sigma}}{\chi},
\tag{35}
$$

with the result from this computation found in Table 3. In this table we also present approximation of the statistical error of the circuit by taking 100 batches of shots ($10^7$ each) and extracting the envelope of the output values. Recall again the excluded factor of $\varepsilon^{-4}$ in the computations.

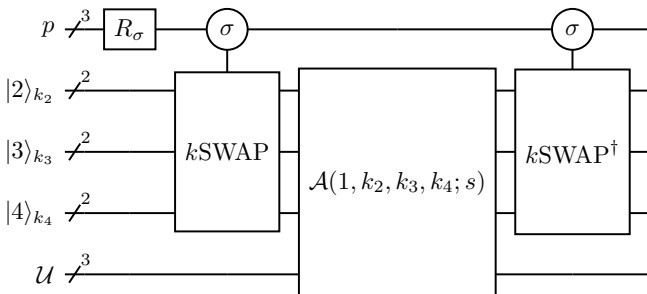

Figure 5: Multi-partial amplitude circuit diagram that performs the computation in Eq. (34).

## 4.3 QFT circuit

We test a simplified version of the full 4-gluon amplitude in the $s$-channel with the test color configuration $\{1, 2, 4, 5\}$ with a focus on the performance of the QFT part of the circuit. This simplified test utilizes the unitarisation method to perform the abstract action of Eq. (16), i.e., we once again start with a normalized superposition of the 6 color-ordered states in the permutation register together with a 2-qubit unitarity register

$$|\Sigma\rangle_p |\Omega\rangle_{\mathcal{U}} = \frac{1}{\sqrt{6}} \sum_\sigma |\sigma\rangle_p |\Omega\rangle_{\mathcal{U}} \tag{36}$$

and then use increment gates $U_+$ and controlled value setting gates $C_{|\sigma\rangle}\big[B(\alpha)\big]$ to apply the corresponding color and helicity factors. The computation goes as follows:

$$\left( \prod_\sigma C_{|\sigma\rangle}\big[B\big(\mathcal{A}(1^-, \sigma(2^-, 3^+, 4^+))\big)\big] \right) U_+ \left( \prod_\sigma C_{|\sigma\rangle}\big[B\big(\mathrm{Tr}\big[T^1 \sigma(T^2, T^4, T^5)\big]\big)\big] \right) U_+ |\Sigma\rangle_p |\Omega\rangle_{\mathcal{U}}$$

$$= \frac{1}{\sqrt{6}} \sum_\sigma \mathrm{Tr}\big[T^1 \sigma(T^2, T^4, T^5)\big] \mathcal{A}(1^-, \sigma(2^-, 3^+, 4^+)) |\sigma\rangle_p |\Omega\rangle_{\mathcal{U}} + \left( \perp |\Omega\rangle_{\mathcal{U}} \right). \tag{37}$$

With this simplified circuit we get a similar output state as in Step 3.2 in the full algorithm and are now ready to use the QFT on register $p$ to get

$$\frac{1}{\sqrt{48}} \sum_\sigma \mathrm{Tr}\big[T^1 \sigma(T^2, T^4, T^5)\big] \mathcal{A}(1^-, \sigma(2^-, 3^+, 4^+)) |\Omega\rangle_p |\Omega\rangle_{\mathcal{U}} + \left( \perp |\Omega\rangle_p |\Omega\rangle_{\mathcal{U}} \right), \tag{38}$$

where the QFT has put the pure summation in front of $|\Omega\rangle_p$ and added a factor of $1/\sqrt{8}$ to the normalization. Since all the partial amplitudes have norm larger than 1, we universally divide them all by a chosen $\varepsilon$ in the input to the $B$ gates which is extracted afterwards. The circuit diagram for this test can be seen in Fig. 6.

| Amplitude | Output value | True value | Relative error |
|-----------|--------------|------------|----------------|
| $\left|\mathcal{A}(1^-,2^-,3^+,4^+)\right|^2$ | $2.04334 \pm 0.065$ | $2.04343$ | $0.004\%$ |
| $\left|\mathcal{A}(1^-,3^+,2^-,4^+)\right|^2$ | $22.63446 \pm 0.195$ | $22.63721$ | $0.012\%$ |
| $\left|\mathcal{A}(1^-,2^-,4^+,3^+)\right|^2$ | $11.08095 \pm 0.135$ | $11.07807$ | $0.026\%$ |
| $\left|\mathcal{A}(1^-,3^+,4^+,2^-)\right|^2$ | $11.0783 \pm 0.128$ | $11.07807$ | $0.002\%$ |
| $\left|\mathcal{A}(1^-,4^+,3^+,2^-)\right|^2$ | $2.04493 \pm 0.057$ | $2.04343$ | $0.074\%$ |
| $\left|\mathcal{A}(1^-,4^+,2^-,3^+)\right|^2$ | $22.63856 \pm 0.206$ | $22.63721$ | $0.006\%$ |

Table 3: Results for the output of Eq. (35). Above the circuit was run 100 times with $\chi = 10^7$ and $\varepsilon = 1.825$ where the average value and the envelopes as statistical error approximations are presented together with the true value comparison.

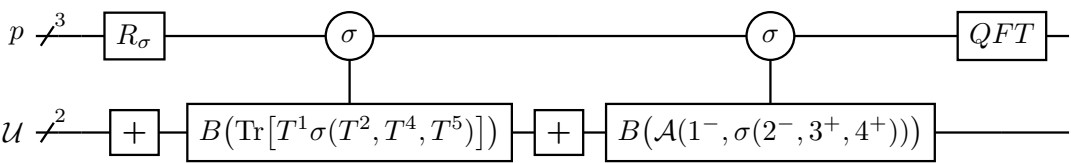

Figure 6: Simplified version of the 4-gluon scattering amplitude algorithm meant to test the efficiency of the QFT operation.

By running this circuit and searching for the probability of reading out the state $|\Omega\rangle_p |\Omega\rangle_\mathcal{U}$ we directly find a value for $\left|\mathcal{M}_{\text{Tree}}(1^-,2^-,3^+,4^+)\right|^2$ (ignoring the coupling constant $g$). The circuit was run 100 times with $\chi = 5 \cdot 10^7$ number of shots each time. The average value found is presented below:

**Output value:** $0.05632 \pm 0.00463$   **True value:** $0.05634$   **Relative error:** $0.028\%$.

This result is promising as it shows that up to a marginal error the QFT performs the expected action. This error rate should be improvable through an increase in shots of the circuit.

## 4.4  Analysis of the $\chi$ and $\varepsilon$ parameters

We study the effects of tuning the shots variable $\chi$ and the unitarity parameter $\varepsilon$ for the multi-partial amplitude circuit in Fig. 5, by investigating how the accuracy of the output varies with these parameters. We introduce 10 distinct spinor sets, reported in Table C.2, to evaluate the dependence of the results on the input values. In the table, we also report what we refer to as optimal $\varepsilon$, which is obtained from computing all the possible spinor products for the given input and evaluating the smallest possible one, according to Eq. (12).

First we show results for the first three sets in Table C.2 for a uniform value of $\varepsilon = 5$ (chosen arbitrarily as a constant value which is larger than the optimal values) and see how relative errors change for a range of shots. The results are presented in the upper plots of Fig. 7. We then choose the optimal $\varepsilon$ for each set separately, which are presented in the first three entries of Table 4 and study again for the same range of shots. The results are seen in the lower plots of Fig. 7 where the relative errors are plotted against the $\chi$ variable for the different kinematic inputs and the various color blobs indicating the different color orders. The error bars in the plot are computed in the same manner as those in Table Table 3, by computing the average and the envelope of 100 batches of independent runs. In both of these tests (upper and lower plots) we show a line which is the average over the color orderings where we can note a general trend of decline in the relative errors with an increasing number of shots. The trend is not always clear for each step as the average oscillates for a set of these plots, however, it is still apparent that the error can be suppressed and convergence be improved by increasing $\chi$. The prime effect comes from optimizing $\varepsilon$, which can be seen on the $y$-axis difference when comparing the upper and lower plots in Fig. 7. The relative errors become substantially smaller when optimizing $\varepsilon$ for each set. The reason for this is likely due to the fact that, when picking the smallest possible $\varepsilon$, the factor $\varepsilon^{-4}$ will not suppress the reference state of interest significantly, making it more likely to measure. If $\varepsilon$ is picked too large, the probability of measuring the state becomes irreparably difficult to find, thus leading to errors of larger magnitude as seen in the upper plots of Fig. 7.

| Spinor set | 1 | 2 | 3 | 4 | 5 | 6 | 7 | 8 | 9 | 10 |
|---|---|---|---|---|---|---|---|---|---|---|
| $\varepsilon$ | 1.825 | 1.754 | 1.422 | 1.669 | 2.293 | 4.937 | 1.769 | 2.412 | 1.636 | 3.808 |

Table 4: Optimal $\varepsilon$ values for the 10 spinor sets tested in Fig. 7 and Fig. 8, obtained by taking the smalllest of all possible combinations of spinors in Eq. (12).

In Fig. 8 we present similarly the relative errors for the 10 different spinor sets found in Table C.2, each with the optimal $\varepsilon$ value, presented in Table 4, with a fixed number of shots. Again, an average (black solid line) is presented over all the spinor sets, for each color ordering. What can be seen in this figure is that some color orderings seem to be more sensitive to the input value (permutation 1 and 5 here), especially for certain spinor sets. This comes as no surprise, as the different color orderings exhibit different singularities between the external particles, being possibly more sensitive on the exact kinematic input values. It should also be noted that the sets with the outlier points in Fig. 8 are sets 6 and 10 which are precisely the sets with the largest $\varepsilon$ value as seen in Table 4 which entail a larger number of shots for improving their accuracy. Lastly we also refer to the fact that the outlier permutations (1 and 5) are at a much smaller magnitude than the rest as seen in Table C.1 in Appendix C hinting at their sensitivity.

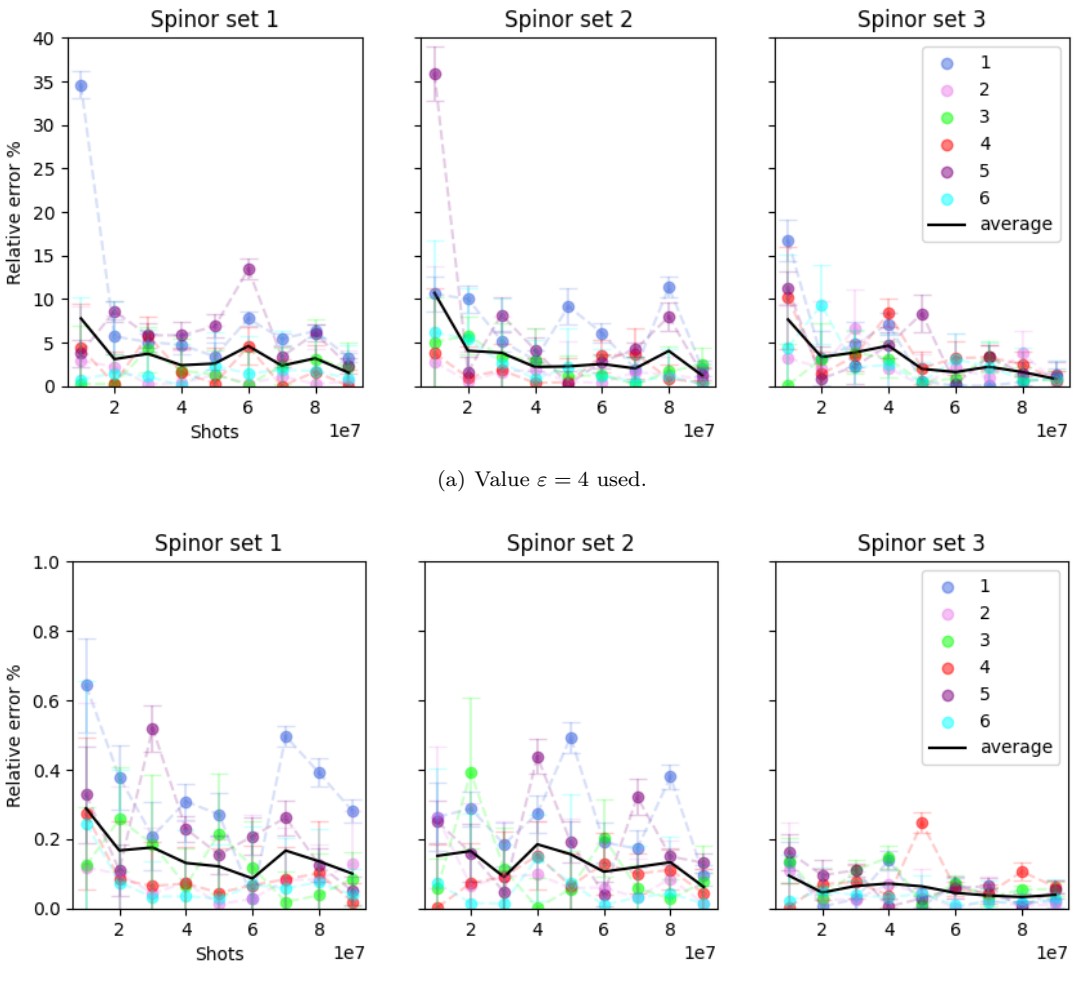

(a) Value $\varepsilon = 4$ used.

(b) Optimized $\varepsilon$ value used, more information in text.

Figure 7: Relative errors of the partial amplitudes labeled $\{1, 2, ..., 6\}$ against the number of shots $\chi$ of the circuit in Fig. 5 for three different spinor value sets found in the appendix in Table C.2. In the upper plots (a), the $\varepsilon$ parameter is uniformly set to $\varepsilon = 4$ for all sets, while the parameter is optimized for each set respectively in the lower plots (b), with values presented in the first three entries of Table 4. The black solid line is a trend line that follows the average over all amplitudes and the labels follow the same order as in Table 3. Error bars with the statistical errors are shown.

## 4.5 Scaling and complexity

In order to properly evaluate the performance of this algorithm we need to carry out a complexity and scaling analysis with increasing $n$. We start with measuring the width of the algorithm, i.e.,

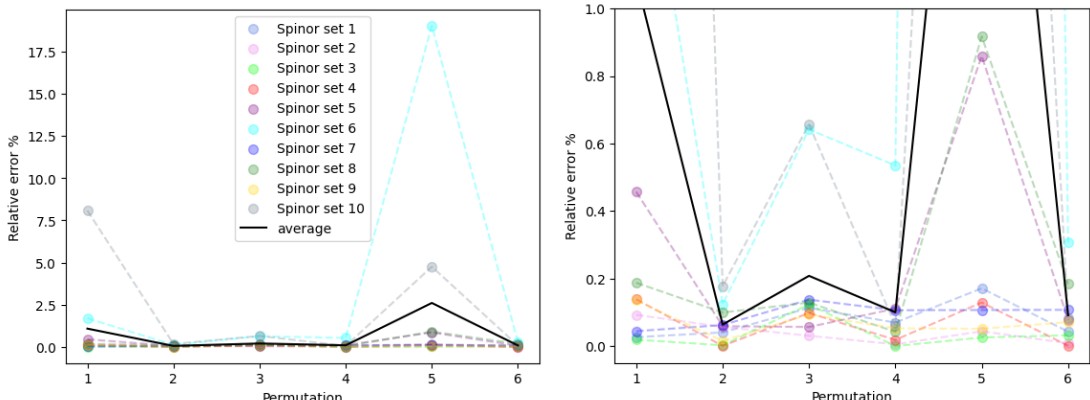

Figure 8: Relative errors for the partial amplitudes labeled $\{1, 2, ..., 6\}$ as presented in Table 3 for 10 different spinor value sets (left). The number of shots was $\chi = 9 \cdot 10^7$ for all runs and $\varepsilon$ was optimized for each set tabled in Table 4. Zoomed in version of left figure (right).

the number of qubits needed for a given $n$. We check this measurement for each individual register in Table 5 where we use #permutations $= (n-1)!$ and also the fact that, if one wants to represent $m$ states in a given register, one needs $\lceil \log_2(m) \rceil$ qubits in order to encode these fully. This gives the number for the registers $h, p, k_i$. Also recall that we need $n-1$ number of $k_i$ registers and $n$ number of $g_i$ registers, so the numbers in the table need to be multiplied by the latter. It is difficult to give a clear analytical expression for the scaling of the total number of qubits, instead, in order to identify the trend, we have chosen to plot the number of qubits compared with different types of scalings in Fig. 9. For this graph we chose #helicities $= n-1$ to symbolize the different choices of $k$ in Eq. (2) and #operations $= 2n+1$ for the $n$ trace factors, $n$ denominator factors and 1 numerator factor in the color and helicity computations.

| Register | Number of qubits |
|----------|------------------|
| $h$ | $n_h = \lceil \log_2(\#\text{helicities}) \rceil$ |
| $p$ | $n_p = \lceil \log_2((n-1)!) \rceil$ |
| $k_i$ | $n_k = \lceil \log_2(n-1) \rceil$ |
| $g_i$ | $n_g = 3$ |
| $q\bar{q}$ | $n_{q\bar{q}} = 4$ |
| $\mathcal{U}$ | $n_u = \lceil \log_2(\#\text{operations} + 1) \rceil$ |

Table 5: Number of qubits needed for each individual register. In the table above $n$ is the number of gluons in the scattering process.

Without detailing the number of layers, we give a rough estimate of the total gate count of each

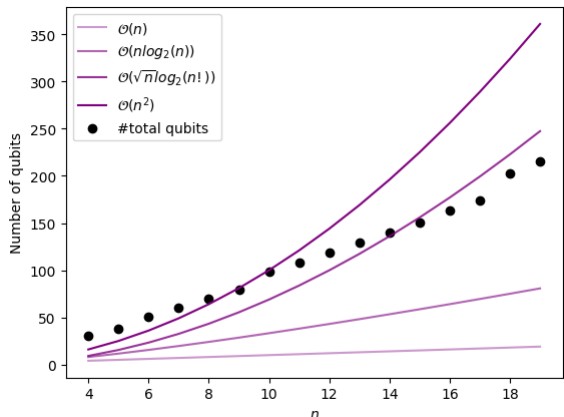

Figure 9: Scaling of total number of qubits for $n$ gluon process compared with possible trends.

step. Also for this measure is difficult to provide a clear value as many gates might have to be decomposed into smaller gates in order to be implementable on hardware. For now, we will assume that these sorts of decompositions only scale linearly for any given gate. If we start with the first layer given in step 1.1, we note that all gates will scale linearly with $n$. For instance, $R_\lambda$ and $R_\sigma$ might need to be decomposed but, by our assumption that the scaling is linear, the Hadamard transformation on the gluon registers will always need $3n$ individual Hadamard gates. The $R_k$ gate is built from $X$ gates that encode the binary representation of the integers $2, ..., n$ and a plot with the number of gates needed can be seen in Appendix C, which also scales linearly. Moving to the controlled-SWAP gates things become large very fast. This is most likely the biggest impediment to the scaling of the algorithm as one will need to implement a large number of controlled-SWAP gates to generate all the desired permutations of the $\{k_i\}$ and $\{g_i\}$ registers. For a rough estimate, this scales factorially and therefore an alternative approach of simplifications might be desirable for larger $n$: we will discuss this in more detail in the outlook part below.

We now study the amount of gates needed to implement a $U_C$ gate. Recall that, in order to build a $U_C$ gate, one strings together $n$ number of $Q$ gates as in Fig. 1, which in turn are built from

$$Q = \big(\Lambda \otimes \mathbb{1}_\mathcal{U}\big)M\big(\mathbb{1}_g \otimes \mathbb{1}_q \otimes U_+\big), \tag{39}$$

where $\Lambda$ has 8 internal gates, $M$ has 17 and $U_+$ is built from $n_u$ controlled-NOT gates (see Ref. [53] for details). However, we can make a simplification of the $M$ gates which yields 7 internal gates rather than 17, see details in Appendix B. Hence, as a rough estimate, the $U_C$ gate scales as

$$\mathcal{O}\big(U_C\big) \sim \mathcal{O}(nQ) \sim \mathcal{O}\big(n(8 + 7 + n_u)\big) \sim \mathcal{O}\big(n\log_2(n_{\text{ops}})\big), \tag{40}$$

where we can approximate further by noting that $n_{\text{ops}} \sim n$. For the $U_\mathcal{A}$ gate on requires a set of $U_+$ gates and a collection of controlled-value setting gates $C_{|ij\rangle}\big[B(\alpha)\big]$ as seen in Fig. 2. For $n$ gluons, one needs $n - 1$ number of $U_+$ gates which in turn are built from $n_u \sim \log_2(n)$ controlled-NOT gates, hence, $\mathcal{O}(n\log_2(n))$. The number of $C_{|ij\rangle}\big[B(\alpha)\big]$ one need is of the order

$$\mathcal{O}\left(n\binom{n-1}{2} + \#\text{helicities}\right) = \mathcal{O}\left(n\frac{(n-1)!}{2!(n-3)!} + \#\text{helicities}\right) \sim \mathcal{O}\left(\sqrt{n}n^2 + n - 1\right) \sim \mathcal{O}(n^{5/2}). \tag{41}$$

Since $\mathcal{O}(n^{5/2}) > \mathcal{O}(n\log_2(n))$ the rough scaling of gates needed for $U_\mathcal{A}$ is $\mathcal{O}(n^{5/2})$. An actual computation of the number of gates needed compared with this scaling can be found in Appendix C.

Moving into the final two regions of Fig. 3 we once again meet the controlled-SWAP gates and $R_k^\dagger$, $R_{q\bar{q}}^\dagger$, which scale as previously explained, however, the final step is the QFT which simply scales as $\mathcal{O}(n_p^2) \sim \mathcal{O}\big(\log_2((n-1)!)\big)$.

Overall, the complete scaling for most of these gates is not particularly bothersome and follows a desirable trend. However, the primary culprits of heavy scaling are the *g*SWAP and *k*SWAP gates.

# 5  Summary

Herein, we have proposed a method for computing the *n*-gluon MHV scattering amplitude at tree-level using QC where we have simulated the specific case $n = 4$. After enforcing a unitarisation procedure of non-unitary operators proposed in literature to construct quantum gates, we have proceeded to design those used for separate color-factor and helicity-amplitude evaluations. When compared to classical calculations, we have found the computation of the former (which are rational numbers emerging from colour algebra in QCD) to be exact and that of the latter (which are irrational numbers stemming from four-momenta configurations in phase space) to be accurate at the below percent level. While computing time is still an issue for the QC approach, we have deemed this results to be encouraging. We have then discussed how these two gates can be combined with a QFT for the computation of the full amplitude, wherein superposition can be utilized to simultaneously sum over all relevant color and helicity configurations. This paves the way then towards a complete implementation of the $n = 4$ gluon amplitude and beyond. One must note however, that with the current state of available QC infrastructure, this algorithm does not outperform classical computations of scattering amplitudes.

While our results are strictly applicable to the $n = 4$ case, we are confident that they serve as a proof-of-principle that can eventually be applied to *n*-gluon tree-level MHV amplitudes with $n \geq 5$. In doing so, though, particular care should be applied to the use of the unitarity parameter $\varepsilon$ and the number of shots $\chi$, which needed to be carefully fine-tuned by hand here to match known results, in turn calling for either a robust self-organising algorithm or else an alternative approach, especially when it comes to the evaluation of unknown MHV amplitudes. Furthermore, one should pay close attention to the scaling of the generalized controlled SWAP gates discussed here, as their number (and in turn that of controlled NOT gates) could potentially become too large to handle in a realistic QC circuit. So that, one may consider some variational transfer method (as proposed in [62]) by initializing the permuted state on some prior circuit and then transfer it to the full algorithm or else adopt an altogether new approach, possibly based on the idea of a Matrix Product State (MPS) [63,64], as used in strongly-correlated many-body quantum systems, which has a strikingly similar structure to that of the scattering amplitude states used here and has already seen implementations on quantum circuits [65,66].

Needless to say, for realistic particle physics applications, one should finally tackle the case of N$^\text{p}$MHV amplitudes as well as the inclusion of (anti)quarks [60,67], so as to have a QC algorithm that encompasses any (massless) QCD amplitude. We leave this endeavor to our future work.

# Acknowledgments

The authors thank Zoltán Trócsányi for his comments on the draft. The authors further thank IBM for the open-source quantum computing platform IBM Quantum and their work on the Python module Qiskit, making this project possible. S. M. is supported in part through the NExT Institute and STFC Consolidated Grant ST/X000583 /1. T.V. is supported by the Swedish Research Council under contract number VR:2023-00221. The computations were enabled by resources within the project UPPMAX 2025/2-312 provided by the National Academic Infrastructure for Supercomputing in Sweden (NAISS).

# A  Unitarisation of non-unitary operators

Here we give a brief review of the unitarisation of non-unitary operators method [53]. Consider the desired action of an arbitrary operator $V$ on some arbitrary quantum state $|\psi\rangle \in \mathcal{H}$:

$$V|\psi\rangle = \alpha|\psi\rangle \quad \text{with} \quad \alpha \in \mathbb{C}. \tag{A.1}$$

The fact that $|\alpha| = 1$ does not hold in general can be circumvented by expanding the Hilbert space, $\mathcal{H} \mapsto \mathcal{H} \otimes \mathcal{H}_\mathcal{U}$ where $\mathcal{H}_\mathcal{U}$ is a *unitarisation space*. For some reference state $|\Omega\rangle_\mathcal{U} \in \mathcal{H}_\mathcal{U}$ an operator $U$ can be considered that acts on $|\psi\rangle \otimes |\Omega\rangle_\mathcal{U} \in \mathcal{H} \otimes \mathcal{H}_\mathcal{U}$ as

$$U\big(|\psi\rangle \otimes |\Omega\rangle_\mathcal{U}\big) = \alpha|\psi\rangle \otimes |\Omega\rangle_\mathcal{U} + \Big( \perp |\Omega\rangle_\mathcal{U} \Big), \tag{A.2}$$

where all the terms to the right are orthogonal to the reference state $|\Omega\rangle_\mathcal{U}$. The expectation value for reading another state $|\phi\rangle \otimes |\Omega\rangle_\mathcal{U}$ is then given by

$$\langle\Omega|_\mathcal{U}\langle\phi|U|\psi\rangle|\Omega\rangle_\mathcal{U} = \langle\phi|V|\psi\rangle \tag{A.3}$$

and thus one has effectively performed the action in Eq. (A.1) by only reading the reference state $|\Omega\rangle_\mathcal{U}$. In a quantum computing setting, the Hilbert space is conventionally expanded by appending an additional register $\mathcal{U}$ which has $n_u$ qubits. A unitary $U$ that acts via Eq. (A.2) can be built from combining a *increment gate* $U_+$ [68] and a *value setting gate* $B(\alpha)$ whose combined action on a state $|k\rangle_\mathcal{U} \in \mathcal{H}_\mathcal{U}$ in an integer representation $k \in \{0, 1, ..., 2^{n_u} - 1\}$ is

$$B(\alpha)U_+|k\rangle_\mathcal{U} = \begin{cases} |0\rangle^{\otimes(n_u-1)} \otimes \big(\alpha|0\rangle + \sqrt{1 - |\alpha|^2}|1\rangle\big), & \text{if } k = 0, \\ |0\rangle^{\otimes(n_u-1)} \otimes \big(\sqrt{1 - |\alpha|^2}|0\rangle - \alpha|1\rangle\big), & \text{if } k = 2^{n_u} - 1, \\ |k+1\rangle_\mathcal{U}, & \text{else.} \end{cases} \tag{A.4}$$

These combine into the desired unitary operator as

$$U = \mathbb{1} \otimes \big(B(\alpha)U_+\big). \tag{A.5}$$

The reason this works is because the increment gate acts as

$$U_+|k\rangle_\mathcal{U} = |k \oplus_{2^{n_u}} 1\rangle_\mathcal{U}, \tag{A.6}$$

where $\oplus_{2^{n_u}}$ is addition modulo $2^{n_u}$ and the value setting gate is built from

$$B(\alpha) = C_{\underbrace{|00...0\rangle}_{n_u-1}}\big[B_1(\alpha)\big], \tag{A.7}$$

where $B_1(\alpha)$ is a single-qubit rotation gate given by the matrix

$$B_1(\alpha) = \begin{pmatrix} \sqrt{1 - |\alpha|^2} & \alpha \\ -\alpha & \sqrt{1 - |\alpha|^2} \end{pmatrix}. \tag{A.8}$$

The notation $C_{|\psi\rangle}\big[U\big]$ means $|\psi\rangle$-controlled-$U$ gate and thus the values setting gate $B(\alpha)$ only applies the single-qubit gate $B_1(\alpha)$ on the first qubit *iff* while all other qubits are in the $|0\rangle$ state, this put together with $U_+$ gives the output in Eq. (A.4). Definitions and circuit representations of these gates can be found in the appendix: in Eq. (B.2) as well as Fig. B.2 and Fig. B.3.

# B  Circuits and operators

The circuit diagram for the quark/anti-quark superposition gate $R_{q\bar{q}}$ which enables Eq. (7) can be seen below in Fig. B.1 where the $R_q$ gate is defined in Eq. (B.1).

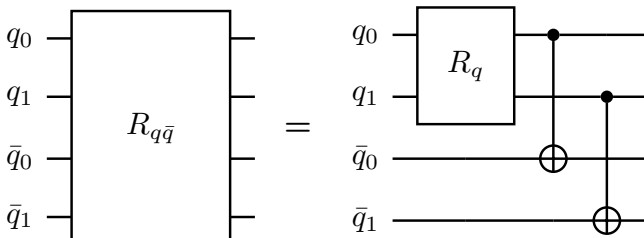

Figure B.1: $R_{q\bar{q}}$ superposition gate which performs the action in Eq. (7) where the $R_q$ unitary is defined in Eq. (B.1).

$$R_q = \begin{pmatrix} \frac{1}{\sqrt{3}} & \frac{1}{\sqrt{2}} & \frac{1}{\sqrt{6}} & 0 \\ \frac{1}{\sqrt{3}} & -\frac{1}{\sqrt{2}} & \frac{1}{\sqrt{6}} & 0 \\ \frac{1}{\sqrt{3}} & 0 & -\sqrt{\frac{2}{3}} & 0 \\ 0 & 0 & 0 & 1 \end{pmatrix}. \tag{B.1}$$

The increment gate $U_+$ is defined as

$$U_+ = \left( \bigotimes_{j=1}^{n_u-1} C_{|1\rangle^{\otimes j}} \left[ X^{(j)} \right] \right) X^{(0)} \tag{B.2}$$

where $C_{|1\rangle^{\otimes j}} \left[ X^{(j)} \right]$ is a multi-controlled CNOT gate and the circuit representation of $U_+$ can be seen in Fig. B.2.

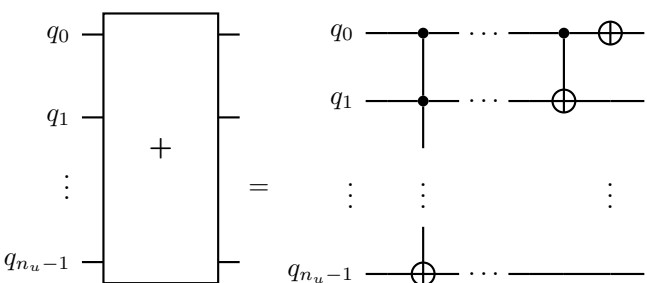

Figure B.2: Circuit diagram of the increment gate $U_+$ that performs the action of Eq. (A.6).

The value setting gate $B(\alpha)$ can be seen in Fig. B.3.

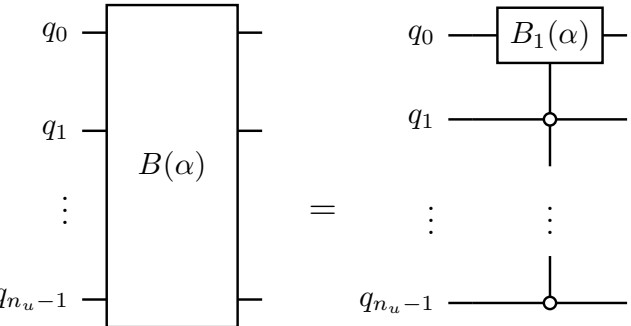

Figure B.3: Circuit representation of the value setting gate in Eq. (A.7).

The $g$SWAP gate is defined by the action

$$g\text{SWAP} \sum_\sigma \frac{1}{\sqrt{n_p}} |\sigma\rangle |a_2 a_3 ... a_n\rangle_{g_2 g_3 ... g_n} = \sum_\sigma \frac{1}{\sqrt{n_p}} |\sigma\rangle |\sigma(a_2, a_3, ..., a_n)\rangle_{g_2 g_3 ... g_n} \qquad (B.3)$$

and a circuit diagram for the simple case $n = 4$ can be seen in Fig. B.4.

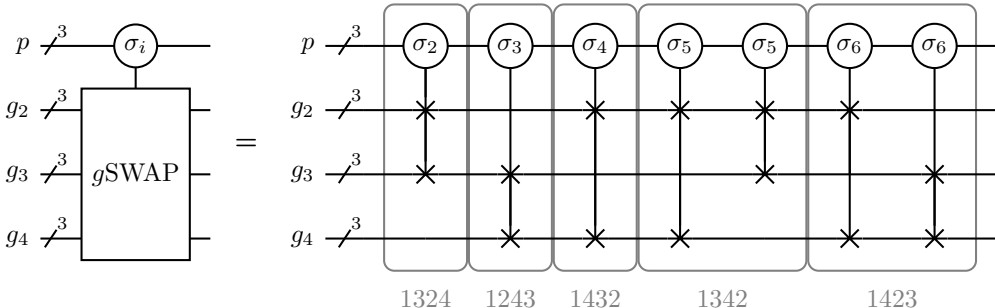

Figure B.4: Circuit diagram for the $g$SWAP gate in Eq. (B.3) for $n = 4$ where there exists 6 relevant permutations. Recall that $\sigma_1$ is non-permuted state so it does not need to be implemented in the circuit.

The permutation superposition gate $R_\sigma$ that enables the creation of a superposition of 6 permutation states, i.e.,

$$R_\sigma |\Omega\rangle_p = \frac{1}{\sqrt{6}} \sum_\sigma |\sigma\rangle \qquad (B.4)$$

is given by the matrix

$$R_\sigma = \begin{pmatrix} \frac{1}{\sqrt{6}} & -\frac{1}{\sqrt{30}} & -\frac{1}{\sqrt{20}} & -\frac{1}{\sqrt{12}} & -\frac{1}{\sqrt{6}} & \frac{1}{\sqrt{2}} & 0 & 0 \\ \frac{1}{\sqrt{6}} & \frac{6}{\sqrt{30}} & 0 & 0 & 0 & 0 & 0 & 0 \\ \frac{1}{\sqrt{6}} & -\frac{1}{\sqrt{30}} & \frac{4}{\sqrt{20}} & 0 & 0 & 0 & 0 & 0 \\ \frac{1}{\sqrt{6}} & -\frac{1}{\sqrt{30}} & -\frac{1}{\sqrt{20}} & \frac{2}{\sqrt{12}} & 0 & 0 & 0 & 0 \\ \frac{1}{\sqrt{6}} & -\frac{1}{\sqrt{30}} & -\frac{1}{\sqrt{20}} & -\frac{1}{\sqrt{12}} & \frac{2}{\sqrt{6}} & 0 & 0 & 0 \\ \frac{1}{\sqrt{6}} & -\frac{1}{\sqrt{30}} & -\frac{1}{\sqrt{20}} & -\frac{1}{\sqrt{12}} & -\frac{1}{\sqrt{6}} & -\frac{1}{\sqrt{2}} & 0 & 0 \\ 0 & 0 & 0 & 0 & 0 & 0 & -\frac{1}{\sqrt{2}} & -\frac{1}{\sqrt{2}} \\ 0 & 0 & 0 & 0 & 0 & 0 & -\frac{1}{\sqrt{2}} & \frac{1}{\sqrt{2}} \end{pmatrix} \tag{B.5}$$

We show a simplification of the $M$ gate defined in Ref. [53] from 17 controlled gates to 7. By noting that the majority of the value settings are $\alpha = 1/2$, we can first append a universal $B(1/2)$ on the $\mathcal{U}$ register which we then reverse for the anomalous entries. These entries are all when $a = 8$ and $k = 1, 2, 3$ and thus we need $1 + 2 \times 3 = 7$ gates. The reduction can be seen below in Fig. B.5.

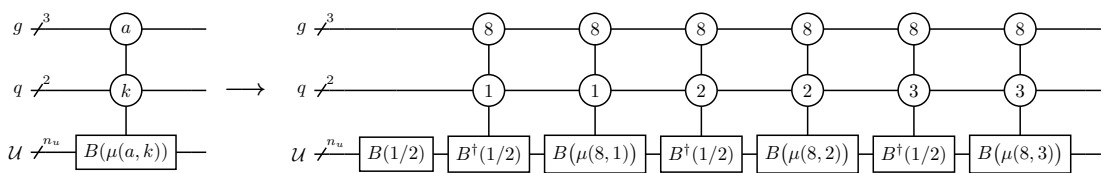

Figure B.5: Simplification of the $M$ gate from 17 to 7 value setting gates.

# C   Tables and plots

The output values for the permuted partial amplitudes for spinor sets 6 and 10 are listed in Table C.1. The full 10 sets of spinors are tabled in Table C.2. The scaling of the number of $X$ gates needed to perform the $R_k$ gate can be seen in the left plot of Fig. C.1. The number of controlled value setting gates needed to construct the $U_\mathcal{A}$ as a function of $n$ is presented in the right plot of in Fig. C.1.

| Spinor set 6 | | | |
|---|---|---|---|
| Amplitude | Output value | True value | Relative error % |
| $\left\|\mathcal{A}(1^-,2^-,3^+,4^+)\right\|^2$ | 1.10589 | 1.08742 | 1.698 |
| $\left\|\mathcal{A}(1^-,3^+,2^-,4^+)\right\|^2$ | 646.47459 | 645.67608 | 0.124 |
| $\left\|\mathcal{A}(1^-,2^-,4^+,3^+)\right\|^2$ | 589.95659 | 593.76835 | 0.642 |
| $\left\|\mathcal{A}(1^-,3^+,4^+,2^-)\right\|^2$ | 590.59189 | 593.76835 | 0.535 |
| $\left\|\mathcal{A}(1^-,4^+,3^+,2^-)\right\|^2$ | 1.29413 | 1.08742 | 19.009 |
| $\left\|\mathcal{A}(1^-,4^+,2^-,3^+)\right\|^2$ | 643.6981 | 645.67608 | 0.306 |
| Spinor set 10 | | | |
| Amplitude | Output value | True value | Relative error % |
| $\left\|\mathcal{A}(1^-,2^-,3^+,4^+)\right\|^2$ | 1.24688 | 1.15374 | 8.073 |
| $\left\|\mathcal{A}(1^-,3^+,2^-,4^+)\right\|^2$ | 241.85933 | 242.289 | 0.177 |
| $\left\|\mathcal{A}(1^-,2^-,4^+,3^+)\right\|^2$ | 211.38299 | 210.00399 | 0.657 |
| $\left\|\mathcal{A}(1^-,3^+,4^+,2^-)\right\|^2$ | 209.87671 | 210.00399 | 0.061 |
| $\left\|\mathcal{A}(1^-,4^+,3^+,2^-)\right\|^2$ | 1.20856 | 1.15374 | 4.752 |
| $\left\|\mathcal{A}(1^-,4^+,2^-,3^+)\right\|^2$ | 242.49014 | 242.289 | 0.083 |

Table C.1: Results for the output of Eq. (35) for spinor sets 6 and 10 with $\chi = 9 \cdot 10^7$ and $\varepsilon = 4.937, 3.808$ respectively.

| Spinor set 1 | | | | |
|---|---|---|---|---|
| Spinor | 1 | 2 | 3 | 4 |
| $\theta_i$ | 0.0 | $\pi$ | 1.326417 | -1.815175 |
| $\varphi_i$ | 0.0 | 0.0 | 1.981338 | 1.160255 |
| Spinor set 2 | | | | |
| Spinor | 1 | 2 | 3 | 4 |
| $\theta_i$ | 0.0 | $\pi$ | 1.214489 | 1.927104 |
| $\varphi_i$ | 0.0 | 0.0 | -2.194071 | 0.947521 |
| Spinor set 3 | | | | |
| Spinor | 1 | 2 | 3 | 4 |
| $\theta_i$ | 0.0 | $\pi$ | 1.580010 | 1.561583 |
| $\varphi_i$ | 0.0 | 0.0 | -0.585922 | 2.555671 |
| Spinor set 4 | | | | |
| Spinor | 1 | 2 | 3 | 4 |
| $\theta_i$ | 0.0 | $\pi$ | 1.285660 | 1.855933 |
| $\varphi_i$ | 0.0 | 0.0 | 0.505077 | -2.636516 |
| Spinor set 5 | | | | |
| Spinor | 1 | 2 | 3 | 4 |
| $\theta_i$ | 0.0 | $\pi$ | 0.902872 | 2.238721 |
| $\varphi_i$ | 0.0 | 0.0 | 0.753240 | 2.388353 |
| Spinor set 6 | | | | |
| Spinor | 1 | 2 | 3 | 4 |
| $\theta_i$ | 0.0 | $\pi$ | 0.407983 | 2.733610 |
| $\varphi_i$ | 0.0 | 0.0 | -2.413538 | 0.728054 |
| Spinor set 7 | | | | |
| Spinor | 1 | 2 | 3 | 4 |
| $\theta_i$ | 0.0 | $\pi$ | 1.202438 | 1.939155 |
| $\varphi_i$ | 0.0 | 0.0 | 2.266519 | -0.875073 |
| Spinor set 8 | | | | |
| Spinor | 1 | 2 | 3 | 4 |
| $\theta_i$ | 0.0 | $\pi$ | 0.855527 | 2.286065 |
| $\varphi_i$ | 0.0 | 0.0 | 2.196657 | -0.944936 |
| Spinor set 9 | | | | |
| Spinor | 1 | 2 | 3 | 4 |
| $\theta_i$ | 0.0 | $\pi$ | 1.316153 | 1.825440 |
| $\varphi_i$ | 0.0 | 0.0 | -2.324677 | 0.816916 |
| Spinor set 10 | | | | |
| Spinor | 1 | 2 | 3 | 4 |
| $\theta_i$ | 0.0 | $\pi$ | 0.531618 | 2.609975 |
| $\varphi_i$ | 0.0 | 0.0 | 1.725324 | -1.416268 |

Table C.2: Spinors sets $\{1, ..., 10\}$ used for $\chi$ and $\varepsilon$ analysis and the 10 phase space point test.

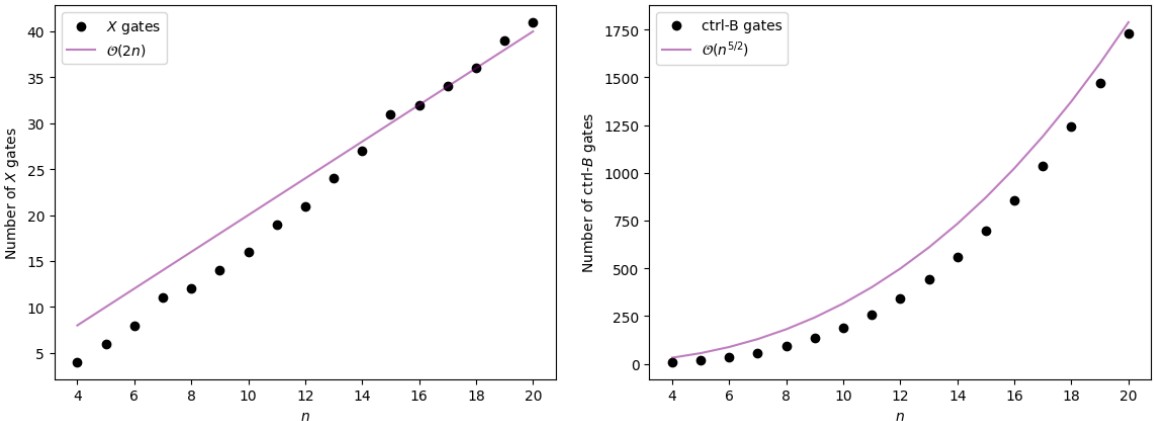

Figure C.1: Number of $X$ gates needed to implement $R_k$ for a given $n$ compared with linear $2n$ scaling (left). Number of $C_{|ij\rangle}\big[B(\alpha)\big]$ gates need for the construction of Fig. 2 compared with $\mathcal{O}(n^{5/2})$ scaling (right).

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
