# Peer review of "A quantum algorithm for the n-gluon MHV scattering amplitude"

_SciPost Physics_

## Round 1 · Referee Report · Anonymous (Referee 1) · 2025-10-13

Report

Dear Editor,

In this manuscript, the authors compute the n-gluon MHV amplitude for n=4 gluons using a quantum algorithm. To that end, they combine two algorithms from the literature to obtain new results. In particular, they use an existing algorithm for the unitarization of non-unitary operations and one for helicity amplitudes. In addition to discussing the proposed new method, they also provide a complexity-scaling analysis for their complete algorithm. The study is performed on a noiseless simulated quantum computer.

The results are particularly interesting and constitute a step further toward obtaining purely quantum simulations of elementary particles at high energies in perturbative computations. This long-term goal might uncover quantum advantages beyond those already known for quantum integration, for example. The work is therefore particularly timely in this emerging field of quantum-computing applications for high-energy physics. The manuscript is well written and clear. Upon addressing the points listed below, I would recommend the manuscript for publication in SciPost.

Main Points

  1. Regarding Eq. (30), it is indicated that the sum over helicity and colour configurations can be performed in post-processing. Would it not be more efficient to perform this summation within the quantum simulation? Are there any limitations in doing so? A discussion addressing this point should be added.

  2. The last paragraph of Section 4.1 is rather unclear, particularly regarding why a measurement cannot be performed and why one should use a state-vector method instead. To clarify this issue, the authors should also explain in a few words what the "state-vector method" is. Additional explanation is needed here.

  3. At the end of Section 4.3 and throughout Section 4.4, the numbers obtained from the quantum simulation should be reported with uncertainties (Gaussian error estimates would be sufficient) related to the number of shots. The number of shots should be provided in every case. This is important, as it is sometimes unclear whether the (dis)agreement with the exact result is due to statistical fluctuations or represents a genuine discrepancy.

Overall, this part of the article lacks clarity. For example, sentences like "We then optimize epsilon for each set separately which are presented in the first three entries of Table 4 and study again for the same range of shots" are very unclear. What are the other entries in the Table for example? In general, it is not clear from the text what method has been used to "optimize" the value of epsilon. Were all possible values checked and then compared? Given that there are statistical fluctuations when performing an experiment with a given number of shots, it is unclear how this optimization can be made meaningful. How was it done in practice? More detailed explanations and reformulations are needed here.

  1. In Figure 7: first, there should be different labels on the upper and lower plots to distinguish them. Otherwise, this can be very confusing. Second, related to the previous point, error bars should be added to each data point to clarify whether differences with respect to the truth represent genuine effects or merely statistical fluctuations.

Minor Points

  1. At the end of the introduction, the statement about Ref. [55] is incorrect. The algorithm is as general as in Ref. [53] and is not restricted to "gluon emissions from a fermion line". The main contribution of Ref. [55] is that it allows the computation of colour amplitudes (and therefore interferences) instead of squared amplitudes as in Ref. [53].

  2. In Figure 9, there is no label on the y-axis. This should be corrected.

Best regards, the referee

Recommendation

Ask for minor revision

---

## Editorial Decision

in_refereeing